# Rapid Adaptation of *Chimonobambusa opienensis* Leaves to Crown–Thinning in Giant Panda Ecological Corridor, Niba Mountain

**DOI:** 10.3390/plants12112109

**Published:** 2023-05-26

**Authors:** Di Fang, Junren Xian, Guopeng Chen, Yuanbin Zhang, Hantang Qin, Xin Fu, Liyang Lin, Yuxuan Ai, Zhanbiao Yang, Xiaoxun Xu, Yuanxiang Yang, Zhang Cheng

**Affiliations:** 1College of Environmental Sciences, Sichuan Agricultural University, Chengdu 611130, China; fangdi3434_cici@163.com (D.F.); qin-hantang@outlook.com (H.Q.); fu__xin@163.com (X.F.); 2394559793@foxmail.com (L.L.); 250516811@foxmail.com (Y.A.); yzb195@126.com (Z.Y.); xuxiaoxu2013@163.com (X.X.); 529877087@foxmail.com (Y.Y.); 29442926@foxmail.com (Z.C.); 2College of Forestry, Gansu Agricultural University, Lanzhou 730070, China; chgp1986@gmail.com; 3Institute of Mountain Hazards and Environment, Chinese Academy of Sciences, Chengdu 610041, China; zhangyb@imde.ac.cn; 4School of Ecological and Environmental Sciences, East China Normal University, Shanghai 201100, China

**Keywords:** *Chimonobambusa opienensis*, leaf traits, trade-off, short-term effect, crown–thinning

## Abstract

Leaf traits reflect the ecological strategy in heterogeneous contexts and are widely used to explore the adaption of plant species to environmental change. However, the knowledge of short-term effect of canopy management on understorey plant leaf traits is still limited. Here, we studied the short-term effect of crown–thinning on the leaf morphological traits of bamboo (*Chimonobambusa opienensis*), an important understorey plant and staple food for the giant panda (*Ailuropoda melanoleuca*) of Niba Mountain. Our treatments were two crown–thinnings (spruce plantation, CS, and deciduous broad-leaved forest, CB) and two controls (broad-leaved forest canopy, FC, and the bamboo grove of clearcutting, BC). The results showed that: the CS enhanced the annual leaf length, width, area, and thickness, CB decreased almost all annual leaf traits, and perennial leaf traits in CS and CB were the opposite. The log-transformed allometric relationships of length vs. width, biomass vs. area were significantly positive while those of specific leaf area vs. thickness were significantly negative, which varied largely in treatments and age. The leaf traits and allometric relationships suggested that the CS created a more suitable habitat for bamboo growth. This study highlighted that the understorey bamboo leaf traits could adapt the improved light environment induced by crown–thinning rapidly.

## 1. Introduction

The leaf is a sensitive organ of plant adaptation to environmental changes. The leaves have been widely explored the adaptative strategies at altitude [1], latitude [2] and success [3] gradients. Plants often improve the performance via adjusting the functional traits (e.g., leaf) to acquire resources [4] for growth and reproduction. This suggests that leaf traits correlate and covary with the other traits [2,5,6], which are often explored by *y* = *β x^α^* (linearized under the form lg (*y*) = lg (*β*) + *α* lg (*x*), *x* and *y* being the determines whether the relationship is isometric (*α* = 1.0) or allometric (*α* ≠ 1)) [5,7,8,9]. This trait-based approaches are now widely explored from organs (e.g., leaf, stems and roots) [10,11] to whole-plant [5] and ecosytem [12] at the scale from local [13] to global [14], and used to study resource acquisition [5], adaptation to environment change and disturbance [15], community assembly [16] and ecosystem function [15,17]. These prior studies also shows that leaf traits vary largely across lineages, life forms [5], ontogenetic stages, size [18], functional groups [19] and environments [16,17,20,21]. Thinning is an essential silvicultural approach to improve light intensity for understorey [22,23], and have potential effects on forest structure (e.g., species richness and composition) and function (e.g., recovery, regeneration and productivity) [24]. However, the trade-offs of understorey leaf morphological traits (leaf traits, hereafter) induced by thinning have not yet been studied.

The Giant Panda Ecological Corridor in the Niba Mountain (GPECN), has been designed and constructed to connect the giant panda populations in the Daxiangling Mountains and the Qionglai Mountain [25]. *Chimonobambusa opienensis* (bamboo, hereafter) is one of the key staple food for the giant panda (*Ailuropoda melanoleuca*) in the GPECN, naturally distributes in the Liangshan Mountain, Daxiangling Mountain and Qionglai Mountain, ranging from 950 m to 2200 m [26]. The delicious new bamboo shoot is the main forest well-being for the local farmers. After decades of recovery, more dense-canopy (coverage > 90%) secondary deciduous broad-leaved forests (DBF) and spruce (*Picea asperata*) plantations (SP), and some dense bamboo groves of clearcutting (BC) have been well developed in the Niba Mountain. However, the closed canopy of DBF and SP limit bamboo to grow, and dense bamboo groves also limit giant pandas to spend time [27]. This suggests that suitable canopy is beneficial to bamboo growth for giant pandas. Hence, in the Autumn of 2018, the crown–thinning of deciduous broad-leaved forest (CB) and spruce plantations (CS) had been implemented to improve giant panda habitat via promoting the understorey bamboo to grow and develop [28]. This provided a good platform for studying the short-term effect of crown–thinning on bamboo leaf trade-off. In the late August of 2020, after in situ survey, the annual and perennial bamboo leaves in the plots of crown–thinning (CB and CS) and controls (BC and deciduous broad-leaved forest canopy, FC) had been sampled, and the leaf traits had been measured. Our objectives were to: (1) determine the short-term effect of crown–thinning on the different-age bamboo leaf traits; and (2) clarify short-term effect of crown–thinning on the trade-off between core leaf traits.

## 2. Results

### 2.1. Variations of Leaf Traits in Age Categories and Treatments

In the GPECN, the bamboo leaf length, width, thickness, LSI, area, biomass and SLA were 125.05 ± 21.46 mm, 20.37 ± 3.71 mm, 0.14 ± 0.03 mm, 6.23 ± 0.99, 19.34 ± 6.59 cm^2^, 0.101 ± 0.040 g and 201.71 ± 49.73 cm^2^ g^−1^, respectively. The GLM analyses (Table 1) showed that the age had significant effects on all traits (all *p* < 0.01); Treatments had minor effect on thickness (*p* = 0. 078), area (*p* = 0.115) and biomass (*p* = 0.088), and significant effects on the other traits (all *p* < 0.01); Treatments × age had significant effect on length (*p* = 0.018), area (*p* = 0.023) and the other traits (all *p* < 0.01), minor effect on LSI (*p* = 0.312), respectively. The age had significant effect on most leaf traits, but had minor effects on the thickness of BC, the SLA of CB and CS, and the LSI of CS (Figure 1).

The boxes of annual leaf traits were showed in Figure 1. The length (154.43 ± 17.26 mm) and area (28.64 ± 5.11 cm^2^) of CS were significant than those of FC (134.40 ± 12.93 mm and 22.34 ± 4.4 cm^2^, respectively) and BC (126.47 ± 19.01 mm and 20.92 ± 5.91 cm^2^). The width (23.90 ± 2.83 mm), thickness (0.145 ± 0.027 mm) and biomass (0.128 ± 0.03 g) of CS were significantly higher than those of FC (20.60 ± 2.43 mm, 0.127 ± 0.017 mm and 0.098 ± 0.022 g, respectively). The LSI of CS (6.49 ± 0.59) was significantly higher than that of BC (5.47 ± 0.69). The CB decreased almost all annual leaf traits significantly. The width, thickness and biomass of BC were significantly higher than those of FC, while the LSI of BC was the opposite. The other leaf traits between different treatments did not have significant difference.

The boxes of perennial leaf traits were showed in Figure 1. The length (131.13 ± 15.70 mm), thickness (0.163 ± 0.027 mm), area (20.94 ± 5.63 cm^2^), and biomass (0.136 ± 0.036 g) of CB were significantly higher than the other treatments, while the SLA (152.06 ± 20.67 cm^2^ g^−1^) was the opposite. The SLA (244.41 ± 45.14 cm^2^ g^−1^) of CS were significantly higher than the other treatments and the thickness (0.127 ± 0.025 mm) was the opposite. The length (122.97 ± 13.78 mm) and area (17.56 ± 4.52 cm^2^) of CS were significantly higher than those of BC (108.90 ± 16.23 mm and 15.15 ± 4.84 cm^2^). The LSI of FC (7.17 ± 1.23) was significantly higher than the other treatments. The length (121.43 ± 12.30 mm), thickness (0.163 ± 0.02 mm) and SLA (208.82 ± 22.58 cm^2^ g^−1^) of FC were significantly higher than those of BC (108.90 ± 16.23mm, 0.146 ± 0.028 mm and 163.56 ± 17.27 cm^2^ g^−1^, respectively), while the biomass (0.077 ± 0.02 g) of FC was significantly lower than that of BC (0.108 ± 0.038 g). The other leaf traits between different treatments did not have significant difference.

### 2.2. Allometric Relationships between Core Leaf Traits

#### 2.2.1. Length vs. Width

The relationships of length vs. width were allometric except for the perennial leaf in FC (*p* = 0.150, Figure 2A,B). Most relationships of length vs. width followed the “diminishing returns” except for the annual leaf in BC, and there existed significant differences between slopes and 1.0 except for annual leaf in BC, and perennial leaf in CS and FC (Table 2). The SCSs and ISEs between treatments were significant except for annual leaf ISE of CS-FC and CB-BC, annual leaf SCS of FC-CB, perennial leaf ISEs of CB-CS, BC-CS and FC-BC, and perennial leaf SCS of CB-CS and BC-FC (Table 2), respectively. The inter-annual allometric relationships varied only largely in FC, and the allometric relationship of annual leaves in FC was significant (*p* = 0.002, Table 3) while that of perennial leaves was slightly (*p* = 0.150). The ISEs and SCSs were significant except for ISEs in FC (*p* = 0.302) and BC (*p* = 0.064).

#### 2.2.2. Biomass vs. Area

All biomass vs. area were significantly allometric relationship (all *p* < 0.001, Figure 2C,D), and only the difference between the slope and 1.0 of annual leaves in CS was significant. The perennial leaves in CB and FC followed the “diminishing returns”, while the others followed the “increasing returns”. The biomass vs. area of annual leaves had no common slope (*p* = 0.044), and the slope in CS was significantly higher than that in CB-BC (Table 3). Based on the common slope of perennial leaves, ISEs (except for CB-BC, and CS-FC, BC-CS) and SCSs (except for FC-BC) were significant. The inter-annual allometric relationships did exist in all treatments significantly, and all ISEs and SCSs were significant except for ISE in CS (*p* = 0.205, Table 3).

#### 2.2.3. SLA vs. Thickness

The relationships of SLA vs. thickness (Figure 2E,F) were significantly allometric except for that of annual leaves in CB and BC, and perennial leaves in BC, respectively. Only the difference between slope and 1.0 of annual leaves in CS and perennial leaves in BC were significant. The SCSs (except for annual leaves of BC-CS, FC-BC and FC-CS) and ISCs (except for annual leaves of BC-CS, and perennial leaves of BC-CB) were significant (Table 2).

The inter-annual allometric relationships (Table 3) varied only largely in CB, and the SLA vs. thickness of annual leaves was not allometric (*p* = 0.272) while that of perennial leaves was significantly allometric (*p* < 0.01). SLA vs. thickness of each treatment had common slopes except for BC (*p* = 0.012), and the slop of annual leaves in BC was significantly higher than that of perennial leaves (*p* < 0.01). The SCSs and ISCs were significantly in CS, CB and FC (all *p* < 0.001), respectively.

## 3. Materials and Methods

### 3.1. Site

The study was carried out in the core of GPECN, which is located in the Sichuan Daxiangling Provincial Nature Reserve (DNR; 102°29′36″–102°52′24″ E, 29°28′33″–29°43′54″ N, 690–3666 m a.s.l.). As the Largest Reintroduction Base for Giant Panda and southern of Giant Panda National Park, the DNR locates in the southwest of Yingjing County of Sichuan Province, and the area is about 29,000 hm^2^. The major soil is yellow-brown soil, which is classified as Alfisols in the soil taxonomy system of China [29]. The climate is humid, and the rainfall is hefty, with up to 200 rainy days per year [30]. The DNR vegetations vary from the evergreen broad-leaved forest (below 1500 m), coniferous forest (1500–2500 m), and coniferous and broad-leaved mixed forest (2100–2600 m) to the coniferous forest (above 2500 m) [30], respectively.

### 3.2. Experimental Design and Survey

In the Autumn of 2018, near to Management and Protection Station of the DNR (29°40′15.960″ N, 102°36′38.592″ E, 2096 m a.s.l.), the crown–thinning experiment had been carried out in the DBF and SP by removing the one-tree crown of every tree, which was equivalent to basal-area removal by about 50% [27] and decreased the canopy coverage to 40–50%. In the later Autumn of 2020, 8 plots (2 m × 2 m) of four treatments (BC, SC, FC and BC), total 32 bamboo plots bad been set and surveyed. The plots were distributed randomly, and the distance between the two plots was more than 20 m for all treatments. The stand and bamboo characteristics of all treatments were listed in the Table 4. After two-year, the canopy coverage of CF and CB increased in some distance (Table 4), and the bamboo growth varied largely in four treatments. Because the sudden outbreak of COVID-19 at the beginning of 2020, the people around DNR tended to pick and cut new shoots rather than went out to work. So, we did not investigate the current-year bamboo in all plots.

### 3.3. Leaf Sampling and Measurement

We defined that the annual leaf was the leaves (not the current year) from annual bamboos and perennial leaf was the leaves (>2 years) from perennial bamboos (>2 years) [31]. Leaf sample collection and traits determination had been carried out: 3 mean leaves from the middle to the top part of mean annual and perennial bamboo were harvested to measure leaf traits (area, length, thickness and width, *n* = 4 treatments × 2 ages × 30 leaves = 240). The individual leaf area was measured using CI-203 Portable Laser Area Meter (1%, CID, Washington, DC, USA), biomass (killed at 105 °C for 15 min and then dried at 85 °C for 48 h; Taisite, WHL45B, Tianjin, China) were measured with an electronic balance (0.0001 g, Zhuojing Experimental Equipment Co., Ltd., BMS, Shanghai, China) [32], and length, thickness and width were measured using a digital vernier caliper (0.01 mm, Deli, model DL91300, Deli Group Co., Ltd., Ningbo, Zhejiang, China), based on which the specific leaf area (SLA, cm^2^ g^−1^) and leaf shape index (LSI = length/width) were calculated.

### 3.4. Statistical Analysis

Before performing analysis, we examined the data for normality and homogeneity of variances, and log-transformed to correct deviations from these assumptions when needed. We examined the differences in leaf traits among treatments, age, and their interactions by General Linear Model (GLM). We compared the traits by Fisher’s least significant difference (LSD) procedure. The statistical analyses were performed using IBM SPSS 20.0 (IBM Corp., Armonk, NY, USA). We created the figures by OriginPro 2019b (OriginLab Corp., Northampton, MA, USA). The tests were considered significant at *p* < 0.05 level. The data were mean ± S.D.

Based on the lg-transformed data, we analyzed trade-offs between core leaf traits of leaf quality vs. photosynthetic-area (biomass vs. area) [8], light acquisition [4] and photosynthetic potential [33] vs. cold resistance (SLA vs. thickness) [34] and leaf shape (length vs. width) [35], respectively. The allometric relationships between leaf traits were analyzed using the standardized major axis method (SMA) with the SMATR Version 2.0 (http://bio.mq.edu.au/research/groups/comparative/SMATR/download.html, accessed on 19 April 2023) [7]. According to Warton et al. (2006) [7] and Chmura et al. (2017) [9], we tested slope differences among compared treatments. The slope values were compared with Sidak correction (when more than two treatments were compared) when significant differences were found in test 1.0. In those cases where the slopes did not differ statistically, we tested for significance of intercept shift in elevation (ISE) and shift along the common slope (SCS).

## 4. Discussion

Our crown–thinning enhanced the annual leaf size (length, width, thickness, biomass and area) in CS largely [22], and had significant negative effect on those in CB. The reasons might be that: the canopy-openess promoted the relative higher understorey plants to grow [36], the stump sprouting increased shading in sometimes [37], and the height of annual bamboos in CB (148 ± 13 cm) were significant shorter than that of the perennial (200 ± 7 cm, *p* < 0.01, unpublished), and the interaction shading of tree and perennial neighbor bamboo affected the leaf heavily [38], which decreased the bamboo photosynthesis [18]. On the contrary, the height of annual bamboo in FC (143 ± 6 cm) were slightly higher than that of the perennial (137 ± 16 cm, *p* = 0.721, unpublished), and could use the scattered light transformed through the canopy [39], which leaded the leaf to be better than that of CB. Furthermore, another complex response might be that bamboo was a typical clone plant, which daughter ramets (i.e., the annual bamboo in our study) could get resources by itself and share from mother ramets [40] (i.e., the perennial bamboo in our study).

In the present study, the SLA change showed that the bamboo leaf economic spectrum shifted from “slow investment-return” of higher light intensity (i.e., BC) and lower light intensity (i.e., FC and annual leaf of CB) to “fast investment-return” of middle light intensity (i.e., CS) [14]. Combined with other different-age leaf traits, we found that moderate canopy (i.e., CS) rather than sunny (i.e., BC) and dense canopy (i.e., FC and annual leaf of CB) would improve bamboo to grow and develop [41]. This is consistent with the moderate disturbance theory in ecological management [42]. Therefore, bamboo leaves have shown obvious adaptation to the light intensity shift, which meant that our canopy management via crown–thinning could improve bamboo growth. In fact, our treatments were a canopy-shading gradient for bamboo (i.e., BC < CB < CS < FC, Table 4). Unlike more experiments [43], bamboo leaf traits of our study showed a unimodal curve with the gradient rather than a linear relationship, which is consistent with the leaf anatomical traits at levels of plant functional group from eastern China [44], and vascular plant leaf area vs. leaf biomass within the Bailongjiang from northwestern China [45].

In the present study, the relationships of lg-transformed length vs. width and biomass vs. area were positively, while that of the SLA vs. thickness was the converse. Furthermore, the trade-off between SLA and leaf thickness suggested that future management shold pay more attention on cold. Furthermore, the trade-offs of annual leaf in CB were shifted largely, and the shifts did not decrease the leaf quality. For instance, the leaf length, thickness, width and area increased within the crown–thinning conditions (esp. CB). The enhanced light improved the habitat for annual bamboo in CB [4,8,33,34,35]. In other words, the bamboo leaf traits were unimodal curves following our crown–thinning light gradients, which was inconsistent with other studies, their functional traits usually were liner-relationship with latitude [46], elevation [43] and enhancing-light [47,48].

In the persent study, the CB stumps are easy to sprout while the CS stumps are difficult to sprout, and the CB has higher understorey plant-species pool than CS. This leaded the CB shading increased largely than that of CS, which hindered the annual bamboo in CB [36,37]. This is why the similar crown–thinning lead to different results, i.e., the successful crown–thinning was CS. Hence, our future management should pay more attention on the contexts of target stand rather than simply canopy-removal. The result was only the early effects of crown–thinning on bamboo leaf traits. And the comprehensive knowledge of crown–thinning on bamboo need more and longer systematic explorations.

## 5. Conclusions

The crown–thinning had obviously influenced bamboo leaf traits, which were high trait- and age-specific. The bamboo leaf could adapt to changing environments quickly, and covaried in traits did not reduce leaf quality. The CS increased light intensity and improved the growth environment for bamboo, and the canopy-removal intensity might appropriately increase aimed to fast-growing and density canopy forests.

## Figures and Tables

**Figure 1 plants-12-02109-f001:**
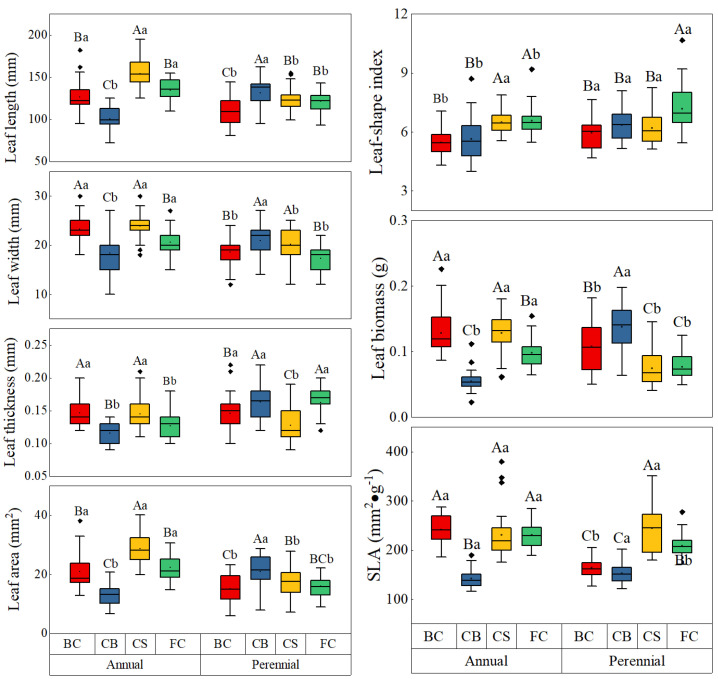
Variations of leaf traits in age categories and treatments. Notes: CS: Crown–thinning of spruce plantation; CB: crown–thinning of deciduous broad-leaved forest; FC: broad-leaved forest canopy; BC: bamboo grove of clearcutting. The same as below. The same capital and lowercase letter were not significantly different (*p* < 0.05) between the treatments and ages.

**Figure 2 plants-12-02109-f002:**
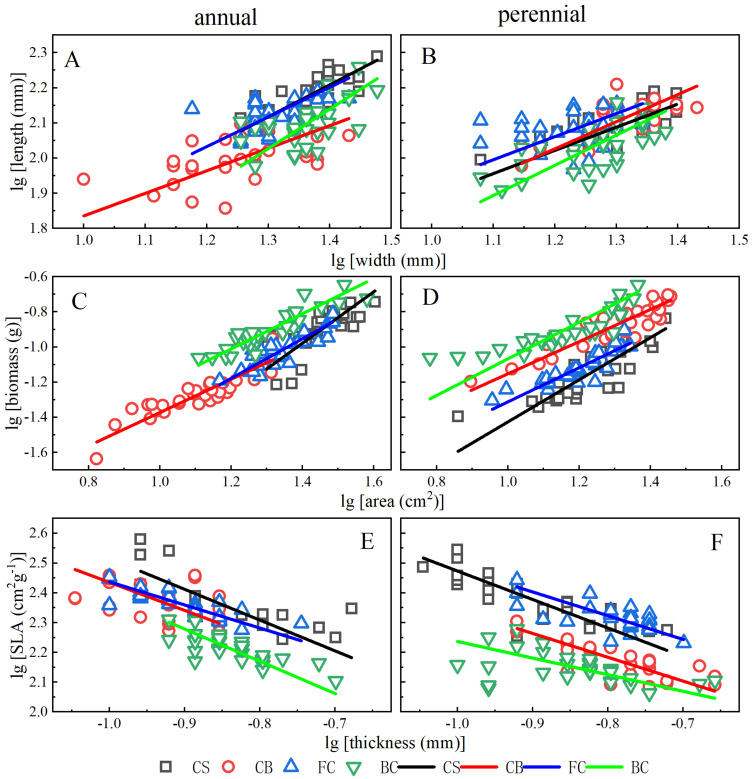
Allometric relationships of length vs. width, biomass vs. area and SLA vs. thickness in age categories and treatments.

**Table 1 plants-12-02109-t001:** The effects (*F*- and *p*-value) of treatments and age categories on leaf traits.

Variance Sources	Treatments		Age		Treatments × Age	
	*F*	*P*	*F*	*P*	*F*	*P*
Length	28.504	<0.001	15.933	<0.001	46.012	<0.001
Width	12.201	<0.001	34.696	<0.001	17.197	<0.001
Area	18.653	<0.001	39.326	<0.001	43.575	<0.001
Thickness	2.394	0.069	30.578	<0.001	26.949	<0.001
Biomass	12.174	<0.001	0.611	0.435	64.173	<0.001
SLA	9.275	<0.001	23.41	<0.001	28.657	<0.001
LSI	20.03	<0.001	10.95	0.001	4.2	0.006

Note: LSI was leaf shape index.

**Table 2 plants-12-02109-t002:** Effects of crown–thinning on the allometric relationships of length vs. width, biomass vs. area and SLA vs. thickness.

Trade-Off	Year	Treatments	*R* ^2^	Slope (95% CI)	Intercept	Common Slope (95% CI)	Intercepts Shift in Elevation			Shifts along the Common Slope		
							CS	CB	FC	CS	CB	FC
Length vs. width	annual	CS	0.494 **	0.910 ^ns^ (0.694, 1.195)	0.934 (0.589, 1.279)	0.874 (0.747, 1.021)	1			1		
		CB	0.266 **	0.642 ** (0.463, 0.889)	1.194 (0.926, 1.462)		**	1		**	1	
		FC	0.291 **	0.844 ^ns^ (0.612, 1.162)	1.020 (0.660, 1.381)		ns	**	1	**	**	1
		BC	0.342 **	1.125 ^ns^ (0.826, 1.533)	0.564 (0.082, 1.046)		**	ns	**	**	**	ns
	perennial	CS	0.392 **	0.660 ** (0.490, 0.888)	1.230 (0.971, 1.490)	0.743 (0.640, 0.863)	1			1		
		CB	0.501 **	0.773 ^ns^ (0.590, 1.013)	1.099 (0.820, 1.377)		ns	1		ns	1	
		FC	0.073 ^ns^	0.657 *(0.456, 0.946)	1.273 (0.971, 1.575)		**	ns	1	*	**	1
		BC	0.434 **	0.858 ^ns^ (0.644, 1.143)	0.95 (0.634, 1.266)		*	**	**	**	**	ns
Biomass vs. area	annual	CS	0.541 **	1.462 ** (1.128, 1.896) ^a^	−3.026 (−3.584, −2.469)							
		CB	0.805 **	0.946 ^ns^ (0.798, 1.122) ^b^	−2.319 (−2.499, −2.140)							
		FC	0.764 **	1.125 ^ns^ (0.933, 1.356) ^ab^	−2.528 (−2.812, −2.243)							
		BC	0.843 **	1.002 ^ns^ (0.860, 1.168) ^b^	−2.214 (−2.416, −2.013)							
	perennial	CS	0.691 **	1.198 ^ns^ (0.967, 1.483)	−2.624 (−2.943, −2.306)	1.014 (0.934, 1.100)	1			1		
		CB	0.824 **	0.915 ^ns^ (0.778, 1.076)	−2.067 (−2.262, −1.873)		**	1		**	1	
		FC	0.786 **	0.973 ^ns^ (0.814, 1.163)	−2.285 (−2.494, −2.077)		**	**	1	ns	**	1
		BC	0.893 **	1.041 ^ns^ (0.917, 1.181)	−2.197 (−2.35, −2.043)		**	ns	**	ns	**	ns
SLA vs. thickness	annual	CS	0.547 **	−1.035 ^ns^ (−1.339, −0.800)	1.481 (1.252, 1.71)	−0.918 (−0.794, −1.064)	1			1		
		CB	0.043 ^ns^	−0.943 ^ns^ (−1.365, −0.651)	1.265 (0.929, 1.601)		**	1		**	1	
		FC	0.66 **	−0.777 *(−0.972, −0.621)	1.66 (1.502, 1.819)		**	**	1	ns	**	1
		BC	0.04 ^ns^	−1.084 ^ns^ (−1.571, −0.748)	1.47 (1.125, 1.816)		ns	**	**	ns	**	ns
	perennial	CS	0.713 **	−0.958 ^ns^ (−1.176, −0.779)	1.516 (1.337, 1.696)	−0.821 (−0.720, −0.932)	1			1		
		CB	0.653 **	−0.801 ^ns^ (−1.004, −0.638)	1.544 (1.398, 1.689)		**	1		**	1	
		FC	0.466 **	−0.798 ^ns^ (−1.055, −0.604)	1.685 (1.506, 1.865)		*	**	1	**	**	1
		BC	0.045 ^ns^	−0.558 ** (−0.808, −0.386)	1.74 (1.561, 1.92)		**	ns	**	**	**	**

Note: BC was bamboo grove of clearcutting. * *p* < 0.05; ** *p* < 0.01; ns, *p* > 0.05. 95% CI were 95% confidence intervals. The superscript labels after *R*^2^ meant the significant-level of allometry relationship, the superscript label after slope meant the significant-level between slope and slope = 1.0, the different superscript lowercase letters after (95% CI) meant that there existed the significant difference (*p* < 0.05) between the slopes, respectively. The same as below.

**Table 3 plants-12-02109-t003:** Effects of age on the allometric relationships of length vs. width, biomass vs. area and SLA vs. thickness.

Trade-Off	Treatments	Common Slope (95% CI)	Intercepts Shift in Elevation	Shifts along the Common Slope
Length vs. width	CS	0.7866 (0.639, 0.966)	**	**
	CB	0.717 (0.582, 0.882)	**	**
	FC	0.7568 (0.593, 0.963)	ns	**
	BC	0.9725 (0.78691, 1.205)	ns	**
Biomass vs. area	CS	1.298 (1.099, 1.536)	ns	**
	CB	0.9297 (0.829, 1.044)	**	**
	FC	1.0422 (0.915, 1.188)	**	**
	BC	1.025 (0.931, 1.128)	**	**
SLA vs. thickness	CS	−0.9868 (−0.842, −1.15862)	**	**
	CB	−0.836 (−0.691, −1.016)	**	**
	FC	−0.7853 (−0.661, −0.9337)	**	**

Note: The SLA vs. thickness in BC had no common slope (*p* = 0.012), and the slope of perennial leaves was significantly higher than that of annual leaves (*p* = 0.01).

**Table 4 plants-12-02109-t004:** Tree-layer characteristics of study plots.

Treatments	Canopy Tree	Average DBH (cm)	Canopy Coverage	Average Height (m)
CB	*Betula* and Litsea	10.21 ± 1.73 a	About 55%	15.44 ± 4.74 a
CS	*Picea asperata*	9.96 ± 1.88 a	About 50%	14.78 ± 4.17 a
FC	*Betula* and Litsea	9.83 ± 1.72 a	About 95%	14.48 ± 4.38 a

Notes: CS: Crown–thinning of spruce plantation; CB: crown–thinning of deciduous broad-leaved forest; FC: broad-leaved forest canopy. Columns (mean ± S.D.) followed by the same letter (s) were not significantly different (*p* < 0.05).

## Data Availability

The data presented in this study are available on request from the corresponding author.

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
