# Peer review of "Rapid Adaptation of Chimonobambusa opienensis Leaves to Crown–Thinning in Giant Panda Ecological Corridor, Niba Mountain"

_plants, 2023, doi:10.3390/plants12112109_

Round 1

Reviewer 1 Report

The work is interesting. This manuscript reports on a study of Rapid adaptation of Chimonobambusa opienensis leaves to crown-thinning in Giant Panda Ecological Corridor, Niba Mountain. The study design meets the general standards and from what I can judge the data is being collected and analyzed appropriately. This work is an unpublished manuscript with relevant information that should be made public in a scientific journal for discussion among scientists working in the field. Authors must be careful with spelling (several errors in the text) and writing. However, some comments should be considered before publishing.

Authors must be careful with spelling (several errors in the text) and writing

Author Response

Thanks very much for your time to review and revise this manuscript. I really appreciate all your comments and suggestions. Those comments are all valuable and very helpful for revising and improving our manuscript. We have studied comments carefully and have made correction which we hope meet with approval. Revised portion were revised in the manuscript. And we checked and updated the references according to MDPI.

Reviewer 2 Report

Dear authors,

This is an interesting and well written manuscript. I really enjoyed reading this research but there are some gaps that not allowed to be accepted in this form.

For example, the Notes: CS: Crown-thinning of spruce plantation; CB: crown-thinning of deciduous broad-leaved forest; FC: 174 broad-leaved forest canopy; BC: bamboo grove of clearcutting, shoyld be present in each table and figure.

Also, in some tables and/or figures there are some letters in bolds.

For the rest, the structure of the manuscript is according to standard guidelines.

Author Response

Thanks very much for your time to review this manuscript. I really appreciate all your comments and suggestions. Those comments are all valuable and very helpful for revising and improving our manuscript. We have studied comments carefully and have made correction which we hope meet with approval. Revised portion were revised in the manuscript. The table 1 did not listed BC, because the table 1 showed tree coverage.  And we checked and updated the references according to MDPI. 

Reviewer 3 Report

Positive notes and comments:

1. The topic of the manuscript is up-to-date as it concerns the rapid adaptation of Chimonobambusa opiennsis Leaves to crown-thinning in Giant Panda Ecological Corridor, Niba Mountain;

2. Introduction and literary reference show the achievements of science in the study area;

3. The scientific study is properly structured and meets the requirement of the scientific journal;

4. The Material and Methods section describes in detail the scientific methodology and is adequate to the working hypothesis;

5. the results are exposed relatively well, with statistical data processing and interesting correlation dependencies have been derived;

6. Although short, a discussion has been made, which includes its own experimental data and the results of similar research related to the environmental state of bamboo forests as a natural habitat for the giant panda;

7. At the end of the manuscript, an exposed conclusion summarizes the most important results of this scientific study.

Negative notes and recommendations:

1. The results shown in Figure 1. regarding Variations of Leaf Traits in Age Categories and Treatments are not visualized well. I recommend an increase in scale by 15-20% for better visualization;

2. Table 3 has incomprehensible fields - for example "Perennial". Please adjust the fields for better visualization;

3. In the discussion section, the authors' erudition is evident. In my opinion, in the manuscript, the discussion should be more detailed and in-depth, which would improve the scientific qualities of the article;

4. Although a conclusion was made at the end of the article, it would be appropriate to make specific practice recommendations and thus the article will become a scientific research, useful for environmentalists, biologists and zoologists engaged in the natural habitat of the giant panda - bamboo forests.

Recommendations to Editors:

The topic of the manuscript is interesting as it concerns the Rapid Adaptation of Chimonobambusa opiennsis Leaves to crown-thinning in Giant Panda Ecological Corridor, Niba Mountain. The article is an extensive scientific and applied research, useful for environmentalists, biologists and zoologists engaged in the natural habitat of the giant panda - bamboo forests. In my opinion, the manuscript has great potential for quotation as it has all the resistance to valuable scientific work. I recommend printing the article after taking into account the notes and recommendations of the reviewers.

Author Response

Thanks very much for your time to review this manuscript. I really appreciate all your comments and suggestions. Those comments are all valuable and very helpful for revising and improving our manuscript. We have studied comments carefully and have made correction which we hope meet with approval. Revised portion were revised in the manuscript. And we checked and updated the references according to MDPI.
